# Venetoclax-Based Regimens in CLL: Immunoglobulin G Levels, Absolute Neutrophil Counts, and Infectious Complications

**DOI:** 10.3390/biomedicines13071609

**Published:** 2025-06-30

**Authors:** Wojciech Szlasa, Monika Kisielewska, Anna Sobczyńska-Konefał, Emilia Jaskuła, Monika Mordak-Domagała, Jacek Kwiatkowski, Katarzyna Tatara, Agnieszka Kuś, Mateusz Sawicki, Izabela Dereń-Wagemann, Mariola Sędzimirska, Ugo Giordano, Jarosław Dybko

**Affiliations:** 1Lower Silesia Centre for Oncology, Pulmonology and Hematology in Wrocław, 53-439 Wrocław, Poland; anna.sobczynska-konefal@dcopih.pl (A.S.-K.); emilia.jaskula@dcopih.pl (E.J.); monika.mordak-domagala@dcopih.pl (M.M.-D.); jacek.kwiatkowski@dcopih.pl (J.K.); katarzyna.tatara@dcopih.pl (K.T.); agnieszka.kus@dcopih.pl (A.K.); mateusz.sawicki@dcopih.pl (M.S.); izabela.deren-wagemann@dcopih.pl (I.D.-W.); mariola.sedzimirska@dcopih.pl (M.S.); ugo.giordano@dcopih.pl (U.G.); jaroslaw.dybko@dcopih.pl (J.D.); 2Department of Molecular and Cellular Biology, Faculty of Pharmacy, Wroclaw Medical University, 50-556 Wrocław, Poland; monika.kisielewska@student.umw.edu.pl; 3Faculty of Medicine, Wroclaw Medical University, 50-556 Wrocław, Poland; 4Hirszfeld Institute of Immunology and Experimental Therapy, Polish Academy of Sciences, 53-114 Wroclaw, Poland; 5Department of Oncology and Hematology, Wroclaw University of Science and Technology, 50-370 Wrocław, Poland

**Keywords:** CLL, venetoclax, rituximab, obinutuzumab, CD20

## Abstract

**Background:** Chronic lymphocytic leukemia (CLL) is a prevalent hematologic malignancy that predominantly affects elderly individuals, posing significant clinical challenges due to patient comorbidities and inherent resistance to conventional chemotherapy. The emergence of targeted therapies combining venetoclax, a selective inhibitor of the anti-apoptotic protein BCL-2, with anti-CD20 monoclonal antibodies has dramatically transformed the treatment landscape. **Methods:** This retrospective observational study analyzed the differential impacts of first-line venetoclax-obinutuzumab (VenO) and second-line venetoclax-rituximab (VenR) on immunoglobulin G (IgG) levels and absolute neutrophil count (ANC) in CLL patients. **Results:** Our findings indicate that during first-line VenO therapy, a significant improvement in ANC levels from baseline was observed, whereas patients undergoing second-line VenR therapy demonstrated limited impact on ANC and the decreasing tendency in IgG levels. Patients treated with VenR had a longer disease history and previous exposure to other treatment regimens, primarily chemoimmunotherapy, which could negatively influence immune recovery, making direct comparisons between these two treatment lines challenging. Although this observational study did not directly compare infection rates, the observed enhancement of ANC levels in patients receiving VenO suggests a potential for lower infection risk compared to pretreated VenR patients. **Conclusions:** These results underscore the clinical significance of considering both the treatment line and the patient’s prior therapeutic history when selecting venetoclax-based regimens for CLL. The potential association of first-line VenO with improved immunological parameters and the complex impact of prior therapies on immunological recovery with second-line VenR warrant further prospective investigation into the correlation between treatment regimen, patient history, immune function, and infectious complications.

## 1. Introduction

Chronic lymphocytic leukemia (CLL) is a prevalent type of hematologic neoplasm, primarily affecting elderly individuals, with a median age of diagnosis at 70 years old. This condition is characterized by the proliferation and accumulation of mature B cells in lymph nodes, blood, and lymphatic tissues [1]. Traditionally managed with chemotherapy, CLL’s treatment has undergone a paradigm shift towards targeted therapeutic approaches, primarily due to significant chemotherapy-associated toxicities and limited efficacy in high-risk patient populations [2]. Despite these advancements, CLL remains considered incurable, due to the potential for patients to develop resistance, which can lead to disease progression and, in some cases, the transformation into Richter Syndrome, an aggressive form of lymphoma marked by diffuse large B-cell lymphoma (DLBCL) [3]. Leukemic transformation in CLL mainly arises from genomic modifications affecting the proliferation and apoptosis regulation of B cells [4]. These transformations usually commence with substantial chromosomal variations, such as deletions in 13q14.3 and 11q, or in 17p and trisomy 12. The most prevalent alteration is the 13q deletion, which is generally associated with a favorable prognosis. In contrast, deletions in 11q and 17p are linked to the worst prognosis, while trisomy 12 deletion represents an intermediate risk [5]. The 17p deletion, frequently accompanied by TP53 mutations, represents a poor prognostic factor in CLL, given TP53’s essential role in cell cycle regulation, DNA repair, and apoptosis, ultimately leading to shorter treatment-free intervals and diminished overall survival [6,7]. In a similar manner, the deletion of 11q, often resulting in the loss of the ATM gene, further impairs p53 functionality, contributing to accelerated disease progression and poor survival outcomes. Both ATM and TP53 are fundamental to maintaining genomic integrity through DNA damage response, and their disruption is associated with genomic instability, aggressive disease progression, and resistance to conventional therapies [3,5,8]. A separate adverse prognostic factor is an unmutated IGHV gene. IGHV mutation status provides insights into the development of B cells, with approximately 60% of CLL cases arising from B cells that have undergone somatic hypermutation within the germinal center. Patients exhibiting unmutated IGHV genes, characterized by ≥98% sequence identity to the germline, demonstrate a more aggressive disease trajectory, reduced progression-free and overall survival, and poorer response to CIT [3,9,10]. These genetic alterations create a challenging clinical scenario, with certain CLL patients showing high resistance to standard therapies, necessitating more aggressive or targeted treatments [11].

The majority of patients with CLL are typically around 70 years old and often have clinically significant comorbidities that render traditional chemotherapy regimens less suitable due to their toxicity. The need for treatment options that are both effective and less harmful was fulfilled by the introduction of novel targeted therapies, such as venetoclax and anti-CD20 monoclonal antibodies [12]. These newer therapies focus on molecular targets essential to leukemic cell survival and proliferation, prominently including venetoclax, a BH3 mimetic inhibiting the critical survival protein BCL-2, and anti-CD20 monoclonal antibodies, namely obinutuzumab and rituximab. The therapy is effective, even in patients with high-risk genetic features [2]. Venetoclax, an orally administered BH3-mimetic, specifically inhibits the anti-apoptotic protein BCL-2, a key regulator that prevents programmed cell death in CLL cells. By binding to BCL-2, venetoclax disrupts its interaction with pro-apoptotic proteins like BAX, thereby promoting apoptosis in CLL cells, which often exhibit overexpression of BCL-2, which leads to resistance to cell death. This mechanism makes venetoclax particularly effective in inducing cell death in CLL [1,3,12,13]. Rituximab and obinutuzumab are both anti-CD20 monoclonal antibodies used to treat B-cell lymphomas and CLL, but they exhibit distinct mechanisms of action and clinical outcomes. Rituximab, a type I antibody, induces cell death mainly through complement-dependent cellular cytotoxicity by clustering CD20 into lipid rafts. Obinutuzumab, a glycoengineered type II antibody, was developed to address rituximab’s limitations by avoiding CD20 clustering and reducing pro-survival signaling, thereby overcoming resistance. Obinutuzumab shows greater affinity to FcγRIIIa, which results in enhanced direct B-cell killing through antibody-dependent cellular cytotoxicity and antibody-dependent cellular phagocytosis [14,15,16]. Clinically, obinutuzumab demonstrates superior efficacy, particularly in patients with previously untreated CLL, where it is associated with longer progression-free survival. However, the increased potency of obinutuzumab can potentially result in higher incidence of adverse events, such as infusion-related reactions or neutropenia, necessitating careful patient selection and management [17,18,19,20,21]. Differences in the therapy outcomes between second-line treatments are related to cumulative immune system damage caused by previous treatments. Additionally, pharmacological differences between rituximab and obinutuzumab impact immune recovery differently, thus influencing therapeutic outcomes [22,23,24].

This retrospective observational study evaluates the differential impacts of VenO and VenR regimens, specifically on immunoglobulin G levels and ANC, key indicators of immune competence, and infection risk. Understanding these hematologic responses provides critical insights into the nuanced benefits and limitations associated with each therapeutic approach.

## 2. Materials and Methods

### 2.1. Study Design

This retrospective observational study analyzed data from patients diagnosed with CLL and treated at the Lower Silesian Oncology Centre from November 2019 to November 2024. Patients were grouped according to their treatment regimens: VenO administered as first-line therapy, and VenR utilized as second-line therapy following previous treatments. The retrospective nature and distinct therapeutic settings precluded direct comparative analyses; thus, each regimen was assessed separately to ensure accuracy in evaluating their individual impact on immunological parameters, specifically IgG and ANC levels. Primary endpoints included the evaluation of changes in IgG and ANC. Secondary assessments included hematological response rates, overall survival, relapse rates, and the impact of genetic mutations.

### 2.2. VenO Patients Group

The VenO patients group included 35 patients (22 males and 13 females) diagnosed with CLL (see Table 1). The average age of the patients was approximately 70.3 years and had the median ECOG score of 0. Fourteen percent of patients had Rai stage 0, twenty nine percent had stage 1, fourty three percent had stage 2, and fourteen percent had stage 4 CLL, with no patients classified as stage 3. Most VenO-treated patients (43%) were diagnosed at Rai stage 2, while a smaller proportion (14% each) were observed at either early (stage 0–1) or advanced (stage 4) stages. The average time from diagnosis to treatment initiation in this group was approximately 158 days. The time to treatment varied widely, from 0 to 690 days, indicating that some treatments were initiated significantly later after diagnosis. The median CLL International Prognostic Index (CLL-IPI) score for this group was 4.39 (SD 2.25).

In this group, IGHV mutated status was observed in ~71% of patients. The interstitial deletion 13q was found in ~17% of patients. Mutations in ATM and TP53 were detected in ~49% of patients. The chromosome 12 trisomy mutation was relatively rare, occurring in ~9% of patients, while the JAK2 mutation was the least frequent, present in only ~6% patients.

The treatment protocol consisted of 12 cycles, each with a duration of 28 days. In cycle 1, obinutuzumab was administered intravenously, beginning with an initial dose of 100 mg on day 1, followed by a dose of 900 mg on day 2, and a dose of 1000 mg on days 8 and 15. Venetoclax was introduced on day 22 of cycle 1 at a starting dose of 20 mg. It was taken orally once daily. In cycle 2, obinutuzumab was continued with a single 1000 mg IV infusion on day 1, while venetoclax dosing was escalated weekly, beginning at 50 mg daily and increasing sequentially to 100 mg, 200 mg, and ultimately 400 mg daily. During cycles 3 to 6, obinutuzumab was administered as a single 1000 mg IV infusion on day 1 of each cycle, with venetoclax maintained at the target dose of 400 mg daily. For cycles 7 to 12, venetoclax was continued at 400 mg daily, with no further administration of obinutuzumab.

In terms of survival and relapse outcomes, 1 patient in this group died, resulting in a mortality rate of 2.86% (1 out of 35 patients). Notably, no relapses were reported during the study period. The baseline laboratory parameters for this group were characterized by a median hemoglobin (HGB) level of 11.61 g/dL, a median platelet (PLT) count of 150 × 10^9^/L, a median leukocyte (LEU) count of 66 × 10^9^/L, and a median lymphocyte count of 7 × 10^9^/L.

### 2.3. VenR Patients Group

The VenR patients group comprised of 16 patients (11 males and 5 females) diagnosed with CLL. The average age of the patients was approximately 71.1 years and had a median ECOG score. Additionally, fourteen percent of patients were classified as Rai stage 0, twenty one percent as stage 1, twenty nine percent as stage 2, twenty nine percent as stage 3, and seven percent as stage 4. Notably, nearly 60% of patients with available data presented with intermediate (stage 2) or advanced (stage 3–4) disease at baseline, reflecting the predominance of higher-risk cases in this group. The average time from diagnosis to treatment initiation in this group was approximately 144 days. The time to treatment varied significantly, ranging from 1 day to 1099 days, indicating that a few cases experienced very delayed treatment. The median CLL-IPI score for this group was 4.86 (SD 2.41). Previous therapies included rituximab (1 patient), R-B (rituximab–bendamustine 5 patients), chlorambucil (4 patients), CVP (2 patients), and R-FC (2 patients). Regarding survival and relapse outcomes, 1 death was reported in the VenR group, indicating a 93.75% survival rate during the study period. The baseline laboratory parameters for this group were characterized by a median hemoglobin level of 12.11 g/dL, a median platelet count of 125 × 10^9^/L, a median leukocyte count of 36 × 10^9^/L, and a median lymphocyte count of 2 × 10^9^/L. In this group, the interstitial deletion of 13q was the most frequent genetic alteration observed, present in ~50% patients. The IGHV mutated status was also notably common, identified in 31% patients. The ATM mutation was identified in ~25% patients, while mutations in TP53, JAK2, and chromosome 12 trisomy were each observed in ~13% patients.

The treatment protocol comprised 24 cycles, each with a duration of 28 days. The regimen commenced with a debulking phase, during which venetoclax was initiated with a 5-week ramp-up, escalating the dose weekly from 20 mg to 400 mg daily by the end of this phase. Beginning with cycle 1, venetoclax was administered daily at a dose of 400 mg, accompanied by rituximab, which was administered intravenously at a dose of 375 mg/m^2^ on day 1. In cycles 2 through 6, rituximab was administered at an increased dose of 500 mg/m^2^ on Day 1, while venetoclax was continued at 400 mg daily. From cycles 7 through 24, venetoclax was maintained at the recommended daily dose of 400 m without the further administration of rituximab. We sustained this regimen until the completion of the 24-month treatment period.

### 2.4. Statistical Analysis

This retrospective observational study evaluates the individual impacts of VenO and VenR regimens on immunoglobulin G levels and ANC, which serve as critical indicators of immune competence and infection risk. The analysis and plots were performed using R studio and appropriate statistical tests. Threshold of the statistical significance was set to *p* < 0.05. It is important to note that no direct statistical comparisons were conducted between the VenO and VenR groups regarding IgG and ANC differences.

## 3. Results

### 3.1. Complete Blood Count Monitoring During the Therapy

The assessment of hematological parameters (Figure 1) in patients treated with VenO confirms the drug’s efficacy over a 12-month period. The median hemoglobin level increased from 11.61 g/dL before treatment to 11.99 g/dL after 3 months, followed by a substantial rise to 13.72 g/dL at 6 months, ultimately reaching 13.89 g/dL at 12 months. This consistent improvement indicates a sustained positive response to therapy, with the most notable increase occurring between 3 and 6 months. Platelets levels also showed a progressive increase, rising from 150 × 10^9^/L before treatment to 162 × 10^9^/L after 3 months, further to 186 × 10^9^/L at 6 months, and reaching 200 × 10^9^/L after 12 months. The most significant increase occurred between 3 and 6 months, suggesting a robust and continuous improvement in platelet counts during treatment. Leukocyte levels dropped sharply from 66 × 10^9^/L before treatment to 7 × 10^9^/L after 3 months, remaining low at 5 × 10^9^/L at 6 months, with a slight increase to 7 × 10^9^/L by 12 months. The significant reduction by 3 months underscores the early effectiveness of the treatment, with subsequent stabilization of leukocyte counts. Lymphocyte levels exhibited a marked decrease from 7 × 10^9^/L before treatment to 3 × 10^9^/L after 3 months, further declining to 1 × 10^9^/L by 6 months, and stabilizing at 2 × 10^9^/L by 12 months. The most pronounced reduction occurred within the first 6 months, demonstrating the therapy’s efficacy in reducing lymphocyte counts.

The analysis of hematological parameters in patients treated with VenR reveals significant changes, indicating the drug’s efficacy over a 24-month period. Hemoglobin levels show a consistent and gradual increase, starting from 12.11 g/dL before treatment and reaching 14.08 g/dL after 24 months. Notable improvements are observed as early as 3 months into the treatment, with significant gains continuing through the 6-month mark. This suggests that rituximab effectively enhances hemoglobin levels, similar to obinutuzumab, but with slightly higher final levels. Platelet levels, on the other hand, demonstrate a more variable response. Initially, PLT levels decrease slightly from 125 × 10^9^/L to 122 × 10^9^/L by 3 months, followed by an increase to 147 × 10^9^/L by 6 months, and then a slight reduction to 141 × 10^9^/L by 24 months. This fluctuation indicates a less consistent effect on platelet counts compared to obinutuzumab, though a notable improvement is observed between 3 and 6 months of therapy. Rituximab’s impact on leukocyte levels is marked by a rapid and sustained reduction. Lymphocytes levels drop sharply from 36 × 10^9^/L before treatment to 5 × 10^9^/L after 3 months, stabilizing around 4 × 10^9^/L through the 6- and 24-month marks. This significant decrease highlights the drug’s quick and consistent efficacy in reducing leukocyte counts, mirroring the effects seen with obinutuzumab but beginning from a lower baseline. Similarly, lymphocyte levels experience a substantial decline, decreasing from 2 × 10^9^/L before treatment to 1 × 10^9^/L within the first 3 months and maintaining this level through 6 and 24 months.

Figure 2 shows a boxplot of the number of symptoms experienced by CLL patients in the VenO group, before and after the treatment. There is a clear reduction in symptoms following therapy, with patients reporting multiple symptoms before treatment and minimal or no symptoms after completing the therapy. The noticeable decrease in symptoms post-therapy indicates substantial clinical improvement.

### 3.2. ANC and IgG Trends During the Therapy

Figure 3A,C demonstrate IgG levels in CLL patients treated with the VenO and VenR regimens across different treatment periods. In the VenO group, two patients received IVIG supplementation during active therapy, and in the VenR group only five patients received IVIG outside the treatment window. Dashed lines on the plot present the IgG trajectories after excluding any patient who ever received IVIG. In VenO, the dashed blue curve still lies below the solid blue curve, confirming that IVIG during therapy attenuates the chemotherapy-induced IgG decline. In VenR group, the dashed red curve is essentially superimposed on the solid red curve, demonstrating that post hoc infusions outside the treatment period have no detectable impact on in-therapy hypogammaglobulinemia.

We also considered neutropenia status. Patients with baseline neutropenia (ANC < 1 × 10^9^/L) were rare at treatment start (2/35 in VenO; none in VenR). However, treatment-emergent neutropenia was more common. In VenO, 4 patients (11%) developed grade 3–4 neutropenia (ANC < 1) during therapy, while in VenR 5 patients (31%) did—likely reflecting cumulative bone marrow damage from prior treatments. Figure 3B illustrates that ANC demonstrated statistically significant stabilization during VenO therapy (*p* = 0.009). After initial variability, neutrophil levels consistently stabilized, implying protective hematologic effects likely associated with reduced infection risks, demonstrating the therapy’s immunological benefit. Patients undergoing second-line VenR therapy (Figure 3D) displayed no significant variation in ANC (*p* = 0.934).

Notably, median baseline ANC in VenO patients who had infections was 2.4 × 10^9^/L vs. 3.5 × 10^9^/L in those without, hinting that a lower neutrophil reserve may contribute to infection risk. By 6 months, ~17% of VenO patients had experienced at least one infection (mostly low-grade), compared to ~63% in VenR. All recorded infections in VenO were mild (CTCAE grade 1), and in VenR two infections of grade 2 severity were documented, with no infections ≥ 3 grade in either group. These infections were predominantly respiratory (e.g., bronchitis) or urinary tract infections, manageable in outpatient settings. No patient required hospitalization for infection, and there were no infection-related deaths. 

### 3.3. Impact of the Lowest IgG Level on the Occurrence of Infections During the Therapy

A total of 6 of 35 patients who received VenO developed infections during the therapy. The mean lowest IgG level in patients who did not develop an infection was approximately 608.5 ± 274.3 mg/dL. In the infected subgroup, the mean lowest IgG level was roughly 470.9 ± 206.0 mg/dL. Although the infected patients exhibited numerically lower mean and median IgG values, the distributions overlapped substantially. A non-parametric Wilcoxon rank-sum test comparing the lowest IgG levels between infected and non-infected individuals yielded a *p*-value above 0.05, thus, no statistically reliable difference in IgG was detected between the two infection subgroups in this cohort.

On the other hand, a total of 10 out of 16 patients who underwent treatment with VenR experienced an infection. Among the non-infected patients, the mean lowest IgG was 304.5 ± 152.7 mg/dL, and for the infected individuals, the mean lowest IgG was 335.4 ± 146.0 mg/dL. The ranges of IgG values for the infected and non-infected subgroups showed substantial overlap. Pearson correlation analysis indicated a very small, non-significant positive association (r ≈ 0.10) between lowest IgG and infection status. Wilcoxon rank-sum testing and an alternative t-test both yielded *p*-values above 0.05 when assessing the difference in IgG levels between those who did and did not experience infections. Thus, no statistically significant difference in IgG was observable for this group. It is worth noting that the effects in VenR patients’ group may be affected by the history of past treatments.

In terms of timing, infections tended to cluster in the early and middle phases of treatment. The median time to first infection in VenO was ~5 months from treatment start (range 1–10 months). For example, 4 patients (11%) in VenO had an infection within the first 28 days (mostly mild respiratory infections coincident with the initial neutropenia nadir), and by 3 months a cumulative 30% had experienced infection. The infection incidence peaked around 3–9 months into VenO therapy, and then tapered off. In VenR, the few infections occurred a bit later, averaging around 6–12 months into therapy.

## 4. Discussion

Hypogammaglobulinemia is a significant complication associated with anti-CD20 monoclonal antibody therapies, such as rituximab and obinutuzumab, in patients with chronic lymphocytic leukemia and other B-cell malignancies or autoimmune diseases [25,26]. Several studies highlight the increased risk of infections in these patients due to lowered immunoglobulin levels [27]. For instance, Athni et al. reported worsening hypogammaglobulinemia following rituximab administration, leading to an overall increase in severe infections [28]. Similarly, the development of post-treatment hypogammaglobulinemia (IgG < 500 mg/dL) was observed in 21% of CLL patients treated with anti-CD20 therapy in one study [27]. The occurrence of late-onset neutropenia has also been documented after treatment with both rituximab and obinutuzumab, with similar incidence rates suggesting a possible class effect for anti-CD20 monoclonal antibodies [29]. However, in this analysis, we observed no link between the occurrence of infection and the lowest IgG level during therapy treatment. Although descriptive findings hinted at minor differences in median IgG between infected and non-infected patients within each therapy protocol group, formal statistical tests did not support the notion that low IgG alone was a primary driver of infection risk. Several factors may explain why IgG levels did not show the existence of a clear relationship with infection. First, other components of the immune system, such as cellular immunity, might play a substantial role in preventing infections in these patient populations. Second, the timing of IgG measurements in relation to infection onset could introduce variability, as IgG can fluctuate over the course of treatment. Additionally, variations in clinical practice—such as prophylactic interventions—might mitigate the impact of low IgG on infection outcomes. These findings underscore the need to consider broader immunologic and clinical factors when evaluating infection risk in individuals receiving VenO or VenR, rather than relying solely on IgG concentration thresholds. Interestingly, other authors consider the management of hypogammaglobulinemia following anti-CD20 therapy, using immunoglobulin replacement therapy (IgRT), either intravenously (IVIg) or subcutaneously (SCIg) [25]. Studies have shown the efficacy and safety of SCIg in secondary immunodeficiency (SID), demonstrating comparable infection prevention to primary immunodeficiency patients, sometimes with lower dosages. A retrospective study also indicated the similar efficacy of facilitated-SCIg in hematological patients with hypogammaglobulinemia compared to IVIg, with a potentially reduced burden on healthcare facilities. Furthermore, the potential for the co-occurrence of hypogammaglobulinemia with other hematological complications, such as anti-CD20 therapy-induced acute thrombocytopenia, as suggested by Haage et al., warrants further investigation. Monitoring immunoglobulin levels before, during, and after anti-CD20 therapy is crucial for the early identification of hypogammaglobulinemia and for guiding appropriate management strategies, including potential IgRT initiation to mitigate the risk of infections [26]. Although the administration of intravenous immunoglobulin was not the primary focus of our analysis, it is likely to have influenced patient outcomes, particularly in those with pronounced hypogammaglobulinemia. By supplementing IgG levels, IVIG can help to compensate for impaired humoral immunity and may reduce the incidence or severity of infections, thereby potentially confounding associations between endogenous IgG levels and clinical events. Additionally, patients receiving IVIG may have been more closely monitored, which could further contribute to improved clinical management and outcomes.

VenO was studied as a fixed-duration therapy, typically administered over 12 months. This regimen demonstrated superior progression-free survival (PFS) compared to chlorambucil–obinutuzumab in patients with untreated CLL and coexisting conditions, as shown in the CLL14 study [22]. VenO achieves high rates of undetectable minimal residual disease, contributing to durable remissions. In terms of safety, VenO is associated with higher rates of neutropenia and infections compared to some other regimens, necessitating careful monitoring [23]. In this study, however, we analyzed the effects of the combination of venetoclax with anti-CD20 antibodies, such as rituximab or obinutuzumab. Preclinical studies have demonstrated that this combination enhances antibody-dependent cellular phagocytosis in an apoptosis-independent manner and improves survival outcomes [30]. It has also been established that this combination effectively promotes apoptosis in MYC/BCL2 co-expressing lymphoma cells and enhances T-cell responses in DLBCL [31]. In clinical trials, VenR therapy has demonstrated considerable efficacy in CLL treatment, as evidenced by the MURANO trial. This trial highlighted the long-term benefits of this regimen, particularly its ability to achieve minimal residual disease negativity and to prolong both progression-free survival (PFS) and overall survival. Notably, even among patients with high-risk genetic markers such as del(17p), TP53 mutations, and unmutated IGHV, those treated with VenR achieved superior outcomes compared to those receiving chemoimmunotherapy, specifically outperforming bendamustine–rituximab in relapsed CLL [32]. However, despite these favorable results, the SARS-CoV-2 pandemic likely contributed to the shorter PFS observed in another study, as the pandemic increased the incidence of adverse events, including neutropenia and infections, complicating the management of CLL patients [33]. Similarly, studies on the VenO regimen have shown remarkable efficacy in previously untreated CLL patients, particularly those with coexisting conditions. Data from the CLL14 trial indicate that this regimen not only achieves a high rate of undetectable MRD but also significantly extends PFS compared to chlorambucil–obinutuzumab. Additionally, patients treated with VenO experienced better quality-of-life outcomes and a lower incidence of second primary malignancies compared to those on chlorambucil–obinutuzumab [22]. In another study, a high discontinuation rate was observed among older patients with poorer baseline renal function, highlighting the need for careful patient selection and management, particularly for those at higher risk of tumor lysis syndrome [34]. Both VenO and VenR have been shown to result in high rates of undetectable MRD and extended PFS, particularly in first-line therapy for unfit patients and those with relapsed CLL. However, venetoclax combined with obinutuzumab has demonstrated superior PFS at three years compared to the combination with rituximab [35]. The results of our study reinforce the established efficacy of combining venetoclax with anti-CD20 antibodies, such as obinutuzumab and rituximab, in the treatment of CLL. Our analysis specifically demonstrated that both obinutuzumab and rituximab, when combined with venetoclax, are effective in managing key hematological parameters in CLL patients. Both regimens led to significant improvements in hemoglobin levels and reductions in leukocyte and lymphocyte counts. Notably, obinutuzumab showed a more consistent increase in platelet levels compared to rituximab, suggesting a potential advantage in managing thrombocytopenia in CLL patients. These findings align with existing clinical evidence that highlights the benefits of venetoclax in combination with anti-CD20 antibodies.

## 5. Conclusions

Patients receiving first-line VenO, in whom improvements in IgG levels and stabilization of neutrophil counts were observed, may potentially experience less hypogammaglobulinemia-related infections compared to patients treated with second-line VenR, where such improvement was not noted. Both groups of CLL patients are at increased risk of infection due to the disease itself and the therapies used, including the risk of late-onset neutropenia associated with anti-CD20 antibodies. Therefore, continuous monitoring for symptoms of infection and regular assessment of immunological parameters, such as IgG levels and neutrophil counts, remain crucial for both treatment groups. 

## Figures and Tables

**Figure 1 biomedicines-13-01609-f001:**
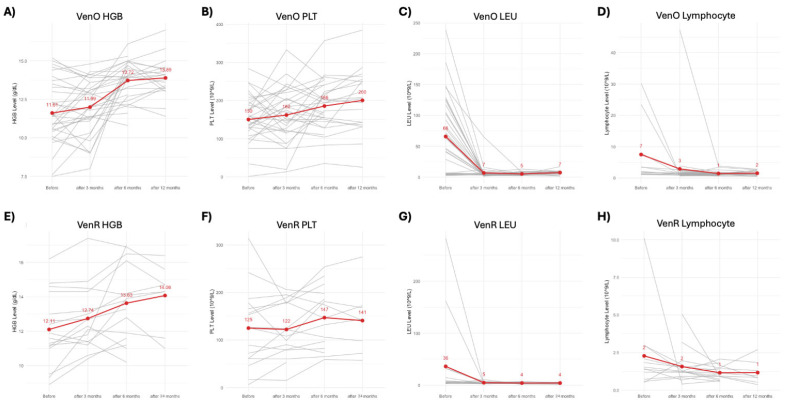
Impact of venetoclax–obinutubumab treatment on hematological parameters in CLL patients over 12 months follow-up: (**A**) hemoglobin (HGB), (**B**) platelet (PLT), (**C**) leukocyte (LEU), and (**D**) lymphocyte levels. Impact of venetoclax–rituximab treatment on hematological parameters in CLL patients over 24 months. (**E**) HGB, (**F**) PLT, (**G**) LEU, (**H**) lymphocyte levels.

**Figure 2 biomedicines-13-01609-f002:**
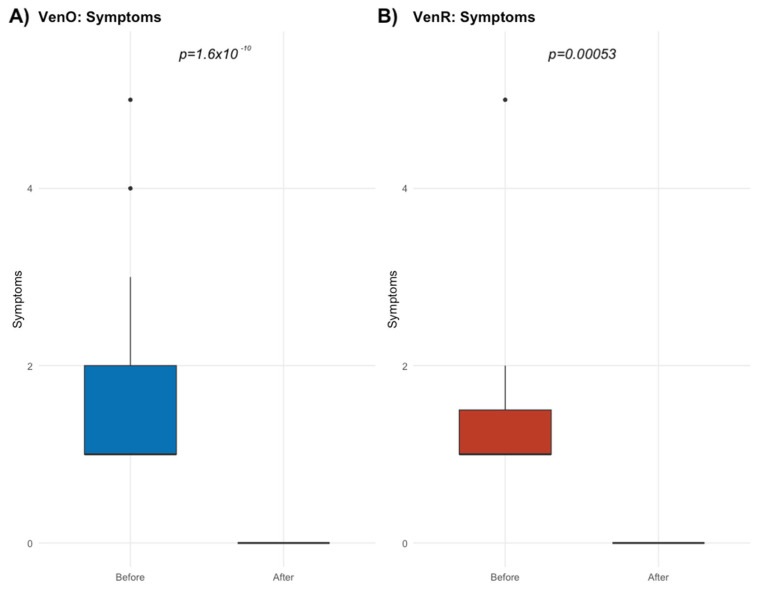
CLL symptom reduction during course of therapy with VenO and VenR; the dots represent outlier observations.

**Figure 3 biomedicines-13-01609-f003:**
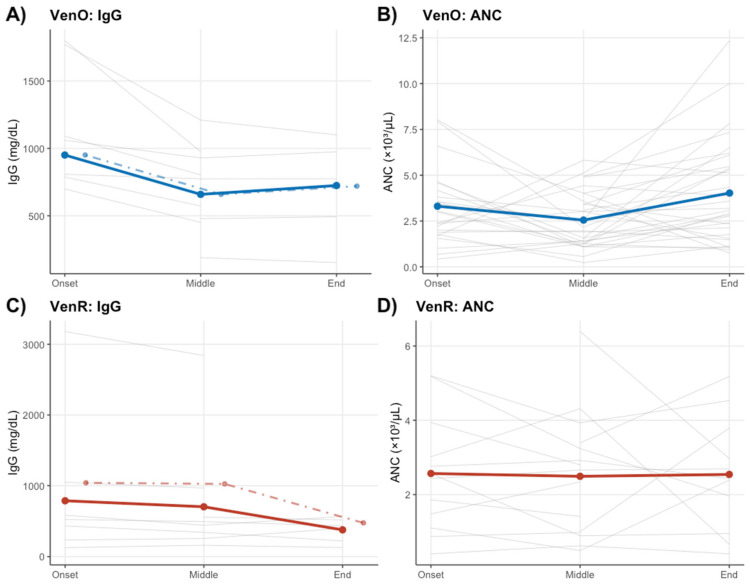
Trends in IgG levels during therapy of CLL with (**A**) VenO and (**C**) VenR. Trends in neutrophil count during the therapy of CLL with (**B**) VenO and (**D**) VenR. The dashed lines represent the group of patients without the ones who received immunoglobulin supplementation (VenO—all during the therapy and VenR—all outside the treatment window).

**Table 1 biomedicines-13-01609-t001:** Characteristics of patients venetoclax–obinutuzumab (VenO) and venetoclax–rituximab (VenR) at the treatment onset.

	VenO Group	VenR Group
Number of Patients	35	16
Gender (M/F)	22/13	11/5
Average Age (years)	70.3	71.1
Average Time to Treatment (days)	158	144
Time to Treatment Range (days)	0–690	1–1099
Median CLL-IPI Score (SD)	4.39 (2.25)	4.86 (2.41)
Deaths	1	0
Median Hemoglobin (g/dL)	11.61	12.11
Median Platelet Count (×10^9^/L)	150	125
Median Leukocyte Count (×10^9^/L)	66	36
Median Lymphocyte Count (×10^9^/L)	7	2

## Data Availability

The original contributions presented in this study are included in the article. Further inquiries can be directed to the corresponding author.

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
