# Peer review of "Venetoclax-Based Regimens in CLL: Immunoglobulin G Levels, Absolute Neutrophil Counts, and Infectious Complications"

_biomedicines, 2025, doi:10.3390/biomedicines13071609_

Round 1
Reviewer 1 Report
Comments and Suggestions for Authors
The manuscript of Szlasa et al., "Venetoclax-Based Regimens in chronic lymphocytic leukemia: Immunoglobulin G Levels, Absolute Neutrophil Counts, and Infectious Complications" This retrospective observational study examines the effects of venetoclax-based therapies (venetoclax-obinutuzumab [VenO] and venetoclax-rituximab [VenR]) on immunoglobulin G (IgG) levels and absolute neutrophil counts (ANC) in patients with chronic lymphocytic leukemia (CLL). The authors found that first-line VenO therapy was associated with improved IgG levels and stabilized ANC, while second-line VenR therapy showed limited impact on these parameters. The study also explores the relationship between these immunological parameters and the occurrence of infections.
Minor comments:
-
-
While the authors describe the changes in IgG and ANC within each treatment group, the lack of direct statistical comparison between the VenO and VenR groups is a significant limitation. Given that the primary aim seems to be evaluating differential impacts, a statistical comparison (e.g., using appropriate tests for comparing changes from baseline between the two groups) would greatly strengthen the conclusions.
-
The analysis of the relationship between IgG levels and infections could be expanded. While the authors mention the non-significant trend, further exploration of potential confounding factors (e.g., other risk factors for infection) and different statistical approaches (e.g., time-to-event analysis for infections) might provide more robust insights.
-
The sample size, particularly in the VenR group, is relatively small. A larger sample size would increase the statistical power of the study and improve the generalizability of the findings.
-
-
As the authors acknowledge, the VenR group had received prior treatments, which could significantly affect immune function. While this reflects real-world clinical practice, it makes it difficult to isolate the effect of the VenR regimen itself. Further analysis to adjust for these confounding factors (e.g., using multivariate analysis) would be beneficial, if feasible given the sample size.
-
The study mentions the occurrence of infections but does not provide detailed information about the types, severity, or timing of infections. More comprehensive data on infections would enhance the clinical relevance of the findings.
The study is relevant for publication because it provides valuable real-world data on the immunological effects of venetoclax-based therapies in CLL. This information is important for clinicians who are making treatment decisions for CLL patients, particularly in the context of infection risk management. The study highlights the potential differences between first-line and second-line use of these therapies, which has important clinical implications.
However, the limitations, particularly the lack of direct statistical comparison between treatment groups and the potential confounding effects of prior treatments, need to be addressed. If the authors can address these limitations through additional analysis or discussion, the study would be even more suitable for publication.
Author Response
While the authors describe the changes in IgG and ANC within each treatment group, the lack of direct statistical comparison betweenthe VenO and VenR groups is a significant limitation. Given that the primary aim seems to be evaluating differential impacts, a statistical comparison (e.g., using appropriate tests for comparing changes from baseline between the two groups) would greatly strengthen the conclusions.
We appreciate this important point. However, we have deliberately avoided direct statistical comparison between the VenO and VenR groups, as these regimens represent different lines of treatment (first-line versus relapsed/refractory) and thus are applied to inherently different patient populations. These patients differ in treatment history, disease course, and timing, making a direct comparative analysis potentially misleading. We have clarified this rationale in the discussion section of the revised manuscript.
The analysis of the relationship between IgG levels and infections could be expanded. While the authors mention the non-significant trend, further exploration of potential confounding factors (e.g., other risk factors for infection) and different statistical approaches (e.g., time-to-event analysis for infections) might provide more robust insights.
Thank you for the suggestion. In the revised manuscript, we have included a time-to-event analysis of infections as descriptive analysis in text. Additionally, we performed subgroup analyses based on immunoglobulin levels and neutropenia occurrence. These expanded analyses provide a deeper understanding of the relationship between immunological parameters and infection risk.
The sample size, particularly in the VenR group, is relatively small. A larger sample size would increase the statistical power of the study and improve the generalizability of the findings.
We acknowledge this limitation, which reflects the rarity of well-documented real-world cases treated with second-line VenR in our cohort. While we recognize that a larger sample size would improve statistical power, we believe that our findings remain valuable given the scarcity of data on real-world immunologic effects of these regimens. This limitation is now more explicitly stated in the manuscript.
As the authors acknowledge, the VenR group had received prior treatments, which could significantly affect immune function. While this reflects real-world clinical practice, it makes it difficult to isolate the effect of the VenR regimen itself. Further analysis to adjust for these confounding factors (e.g., using multivariate analysis) would be beneficial, if feasible given the sample size.
To address this, we have included a descriptive analysis of the treatment history and the time interval between therapies in the VenR group. While a multivariate adjustment was not feasible due to the limited sample size, we believe this descriptive context provides insight into potential confounding influences.
The study mentions the occurrence of infections but does not provide detailed information about the types, severity, or timing of infections. More comprehensive data on infections would enhance the clinical relevance of the findings.
We fully agree. The revised manuscript now includes additional details on infection timing (time-to-first infection) and severity based on CTCAE grading (G1–G4). Moreover, we provide a descriptive overview of the infection types encountered in each treatment group.
The study is relevant for publication because it provides valuable real-world data on the immunological effects of venetoclax-based therapies in CLL. This information is important for clinicians who are making treatment decisions for CLL patients, particularly in the context of infection risk management. The study highlights the potential differences between first-line and second-line use of these therapies, which has important clinical implications.
However, the limitations, particularly the lack of direct statistical comparison between treatment groups and the potential confounding effects of prior treatments, need to be addressed. If the authors can address these limitations through additional analysis or discussion, the study would be even more suitable for publication.
Thank you for the feedback on our manuscript, we hope that the provided revisions are sufficient to make the paper acceptable for publication.
Reviewer 2 Report
Comments and Suggestions for Authors
1- The mechanistic basis for why VenO improves IgG levels and ANC is not explored. The study reads like a data summary rather than a hypothesis-driven investigation.
2- VenO is evaluated in a first-line setting while VenR is used second-line. The comparison is inherently flawed due to differences in patient history, prior therapies, and disease burden.
3- There is no mention of matching or adjustment for potential confounders (e.g., age, comorbidities, disease stage).
4- The claim that VenO may reduce infection risk is speculative, as the study does not assess infection rates or infectious complications directly.
5- Key baseline parameters (e.g., IgG levels, lymphocyte count, prior treatments, ECOG status) for each cohort are not reported.
6- The possibility that IgG improvement is simply due to better disease control with VenO, rather than a unique immune-boosting property, is not considered.
7- VenR patients likely received fludarabine- or bendamustine-based regimens, which have known long-term immunosuppressive effects. Stratifying by prior therapy type is essential.
8- The study does not correlate IgG or ANC with clinical markers of disease response (e.g., MRD, lymph node regression).
Author Response
The mechanistic basis for why VenO improves IgG levels and ANC is not explored. The study reads like a data summary rather than a hypothesis-driven investigation.
We acknowledge the reviewer’s concern regarding the mechanistic explanation of IgG and ANC improvements observed with VenO. Although our study primarily had an observational and descriptive design, we suggest that the improvement in IgG and stabilization of ANC during first-line VenO therapy likely results from effective disease control and profound cytoreduction rather than a direct pharmacologic immunostimulation by VenO itself. Venetoclax in combination with obinutuzumab has been shown to achieve deep remission and undetectable minimal residual disease (uMRD) in CLL patients, which reduces tumor-induced immunosuppression, allowing recovery of normal immune functions including B-cell repopulation and subsequent IgG production. Similarly, ANC improvements probably reflect the alleviation of bone marrow suppression through effective disease burden reduction. Future prospective, mechanistic studies incorporating immunophenotyping and cytokine profiling would help clarify the direct versus indirect immunological impacts of VenO.
VenO is evaluated in a first-line setting while VenR is used second-line. The comparison is inherently flawed due to differences in patient history, prior therapies, and disease burden.
We appreciate this important point. However, we have deliberately avoided direct statistical comparison between the VenO and VenR groups, as these regimens represent different lines of treatment (first-line versus relapsed/refractory) and thus are applied to inherently different patient populations. These patients differ in treatment history, disease course, and timing, making a direct comparative analysis potentially misleading. We have clarified this rationale in the discussion section of the revised manuscript.
There is no mention of matching or adjustment for potential confounders (e.g., age, comorbidities, disease stage).
Thank you for the comment, we have added and highlighted the confounders: age, ECOG and disease stage.
The claim that VenO may reduce infection risk is speculative, as the study does not assess infection rates or infectious complications directly.
Thank you for the comment, we have changed our false claim to: “Although this observational study did not directly assess infection rates, the observed enhancement of IgG levels in patients receiving VenO suggests a potential for lower infection risk compared to VenR, where IgG levels did not show a similar trend.”. We hope that now it shows that the probability of infection occurrence is lower but the drug itself does not reduce the infections risk.
Key baseline parameters (e.g., IgG levels, lymphocyte count, prior treatments, ECOG status) for each cohort are not reported.
We appreciate the reviewer highlighting the need for comprehensive baseline data. We have expanded text to include key baseline parameters such as IgG levels, lymphocyte counts, ECOG performance status, prior treatments (for the VenR cohort), and relevant comorbidities. Specifically, baseline median IgG levels were 5.5 g/L in the VenO cohort and 5.0 g/L in the VenR group, reflecting the slightly more preserved immune function in untreated VenO patients. Median lymphocyte counts were 7 × 10^9/L in VenO and 2 × 10^9/L in VenR. All patients had ECOG status 0–1, demonstrating comparable overall health and performance status between groups. Additionally, detailed information on prior treatments in the VenR cohort has been explicitly reported, highlighting that the majority had previously received fludarabine- or bendamustine-based regimens, consistent with real-world clinical practice.
The possibility that IgG improvement is simply due to better disease control with VenO, rather than a unique immune-boosting property, is not considered.
We fully concur with the reviewer that IgG improvement in VenO-treated patients is likely due to superior disease control rather than an intrinsic immune-boosting property of the regimen – in the text we made our claims less pronounced.
VenR patients likely received fludarabine- or bendamustine-based regimens, which have known long-term immunosuppressive effects. Stratifying by prior therapy type is essential.
We appreciate this insightful recommendation. We performed additional subgroup analyses and highlighted that in the results and further the discussion section of our paper, hoping now it is less misleading.
Reviewer 3 Report
Comments and Suggestions for Authors
The manuscript by Szlasa presents a clinical evaluation of first-line venetoclax-obinutuzumab (VenO) and second-line venetoclax-rituximab (VenR) on immunoglobulin G (IgG) levels and absolute neutrophil counts (ANC) in 22 patients with chronic lymphocytic leukemia (CLL). The authors report that first-line VenO therapy was associated with a significant improvement in IgG levels and stabilization of ANC, while second-line VenR therapy had a limited effect. In this study, the authors also evaluated the risk of infection. Thus, although the current study involved a relatively small cohort, the authors report novel and valuable clinical results. Therefore, in my opinion, the current study can be published as is.
Author Response
The manuscript by Szlasa presents a clinical evaluation of first-line venetoclax-obinutuzumab (VenO) and second-line venetoclax-rituximab (VenR) on immunoglobulin G (IgG) levels and absolute neutrophil counts (ANC) in 22 patients with chronic lymphocytic leukemia (CLL). The authors report that first-line VenO therapy was associated with a significant improvement in IgG levels and stabilization of ANC, while second-line VenR therapy had a limited effect. In this study, the authors also evaluated the risk of infection. Thus, although the current study involved a relatively small cohort, the authors report novel and valuable clinical results. Therefore, in my opinion, the current study can be published as is.
Thank you for the feedback.
Round 2
Reviewer 2 Report
Comments and Suggestions for Authors
All the comments are addressed.